# Magnetic Nanostructures and Stem Cells for Regenerative Medicine, Application in Liver Diseases

**DOI:** 10.3390/ijms24119293

**Published:** 2023-05-26

**Authors:** Tatiane Barreto da Silva, Evellyn Araújo Dias, Liana Monteiro da Fonseca Cardoso, Jaciara Fernanda Gomes Gama, Luiz Anastácio Alves, Andrea Henriques-Pons

**Affiliations:** 1Laboratory of Cellular Communication, Oswaldo Cruz Institute, Oswaldo Cruz Foundation, Rio de Janeiro 21045-900, Brazil; 2Laboratory of Innovations in Therapies, Education, and Bioproducts, Oswaldo Cruz Institute, Oswaldo Cruz Foundation, Rio de Janeiro 21041-361, Brazil; andreah@ioc.fiocruz.br

**Keywords:** liver diseases, magnetic nanoparticles, regenerative medicine, biomaterials

## Abstract

The term “liver disease” refers to any hepatic condition that leads to tissue damage or altered hepatic function and can be induced by virus infections, autoimmunity, inherited genetic mutations, high consumption of alcohol or drugs, fat accumulation, and cancer. Some types of liver diseases are becoming more frequent worldwide. This can be related to increasing rates of obesity in developed countries, diet changes, higher alcohol intake, and even the coronavirus disease 2019 (COVID-19) pandemic was associated with increased liver disease-related deaths. Although the liver can regenerate, in cases of chronic damage or extensive fibrosis, the recovery of tissue mass is impossible, and a liver transplant is indicated. Because of reduced organ availability, it is necessary to search for alternative bioengineered solutions aiming for a cure or increased life expectancy while a transplant is not possible. Therefore, several groups were studying the possibility of stem cells transplantation as a therapeutic alternative since it is a promising strategy in regenerative medicine for treating various diseases. At the same time, nanotechnological advances can contribute to specifically targeting transplanted cells to injured sites using magnetic nanoparticles. In this review, we summarize multiple magnetic nanostructure-based strategies that are promising for treating liver diseases.

## 1. Introduction

Considering the reduced number of livers available for transplantation to those who reach irreversible stages of acute or chronic liver disease, multiple efforts are being made to study alternative therapies. These include cell transplantation, aiming to give these patients a better life quality until transplantation is possible or even natural regeneration of liver function is reached, in some cases [1,2]. Since there are limited sources of liver cells, this approach is also not possible for wide clinical application. The use of stem cells in treating liver diseases is gaining momentum. With the potential for self-renewal and differentiation plasticity, hepatic and extrahepatic stem cells are attracting much attention in regenerative medicine. Among them, multipotent stem cells are the most widely studied [3,4]. However, some challenges must be resolved before stem cells can be used in routine clinical practice. These include the best stem cell source, the optimal route for stem cell transplantation, in addition to the dose and frequency for stem cell administration [5,6,7]. Concerning the ideal infusion way, the most common stem cell transplantation routes include the peripheral vein, the portal vein, and the hepatic artery [8,9]. However, all these approchesimpose some risks to the patient. Among them, the infused stem cells may be trapped in the lungs following peripheral intravenous injection, reducing the number of viable stem cells homing to the target organ, which may decrease treatment efficacy. Direct administration of stem cells into the vessels supplying the liver (the portal vein or the hepatic artery) may also carry substantial risks, such as bleeding and thrombosis [7,10,11]. Therefore, new strategies for delivering stem cells to the site of liver injury could represent major advances in cell transplantation.

Several areas of medicine recognized nanotechnology as an important strategy in drug targeting in addition to treating and diagnosing diseases (e.g., cancer and cirrhosis) [12,13]. Thus, producing magnetic nanoparticles makes it possible to create a controllable means of local-specific cell targeting [14]. This technology can also minimize the alreadymentioned side effects after delivering transplanted stem cells to treat liver diseases.

In this review, we will address the importance of nanotechnology, focusing on the production of magnetic nanostructures targeting stem cells in the treatment of loss of liver function.

## 2. Bioengineering and Stem Cells

Bioengineering is the application of engineering knowledge to biology and medicine, resulting in innovations that can be applied in clinical practice. Regenerative medicine is a bioengineering branch that endows technological developments aiming to restore the functions of an affected tissue or organ [15]. These include cellular therapy when the regenerative capacity of a target tissue is meant to be restored. Different types of cells can be used, among which, stem cells are highly promising and extensively studied in this field owing to their unique characteristics.

Stem cells can be classified as totipotent, pluripotent, or multipotent according to their anatomical origin and capacity to differentiate into other cell types. Totipotent stem cells are found early in embryo development and can differentiate into all cell types and extraembryonic tissues. Pluripotent cells can be isolated from blastocysts or the umbilical cord immediately after birth, and they differentiate into all tissue cells from the three germ layers [16], but not extraembryonic structures [17]. However, the disadvantages of using these stem cells in regenerative medicine include the high risk of rejection and ethical issues when the cells are from embryos.

Multipotent stem cells are isolated from adult tissues and have no ethical concerns involved. These include hematopoietic (HSC), mesenchymal (MSC), neural stem cells (NSC), and others. Multipotent stem cells can be isolated from each patient without relevant rejection risk and be expanded in vitro for bioengineered transplants. However, these cells have reduced plasticity and can only differentiate into specialized tissue-specific cell types. MSCs receive special attention among multipotent stem cells, as they are present in virtually all tissues and have a potent self-renewal capacity [18]. MSCs also affect the tissue ambient by the paracrine secretion of numerous biofactors in vivo, including the induction of other stem cell differentiation [19]. In vitro, the culture media supernatant is named secretome and contains soluble molecules and extracellular vesicles that retain potent biological functions for tissue regeneration [20].

The ideal cellular population best suited for stem cell-based bioengineering should combine the high plasticity of embryonic stem cells (ESCs) and the convenient isolation from patients under treatment. In this regard, induced pluripotent stem (iPS) cells are good candidates. In 2006, a pioneering work showed that the genetic reprogrammation of mouse embryonic and adult fibroblasts led the cells into a pluripotent state, and the authors coined the term “iPS cells” [21]. These cells were generated by using a retrovirus-based gene transfer system carrying the octamer-binding transcription factor 4(Oct3/4), sex-determining region Y-box 2 (Sox2), Krüppel-like factor 4 (Klf4), and c-Myc transcription factors. IPS cell technology brings great promise to medicine, such as personalized cell therapy, disease modeling, and a platform for new drug development and screening. However, iPS cell-based regenerative therapies must circumvent challenges, such as chromosomal instability, reprogramming efficiency, insertional-derived tumor development, and teratoma formation.

Some promising applications include stem cells isolated from the intestines to create in vitro “mini-gut” organoids to treat some digestive disorders [22]. For spinal cord injuries, it was demonstrated in rodents that neural progenitor cells induced axon regeneration and built a new synaptic network [23]. This synaptic structure is scalable to human spinal cord size and lesion geometries, and a 3D bioprinter was used to retrieve spinal cord function. In another example, a significant area of corneal tissue regeneration was achieved using a scaffold designed to encapsulate limbal progenitor/stem cells [24].

In the case of the liver, it is known that the organ has a remarkable capacity to regenerate and recover its average mass after hepatectomy, for example. In this case, parenchymal cells start a first wave of mitosis, followed by the other cell types in a process dependent on extracellular matrix (ECM) remodeling, growth factors, cytokines, and several signaling pathways [25]. However, in cases of chronic damage or extensive fibrosis, for example, the recovery of tissue mass is hampered, and a liver transplant is indicated. The patients could then benefit from alternative bioengineered solutions to improve liver function.

## 3. Stem Cells as an Alternative Treatment to Liver Diseases

The liver is a complex and essential organ due to its multifunctionality, having a role in numerous physiological processes of vital importance, such as the metabolism of macronutrients, protein synthesis and secretion, and drug detoxification [26]. Liver diseases can be caused by numerous factors, such as alcohol and drug abuse, viral infections, and an unbalanced diet, leading to millions of deaths worldwide every year [27,28]. The only definitive treatment in most cases is liver transplantation, which is often not possible because of the low availability of organs compared to the growing demand for transplants [29]. New approaches, such as cell therapy, are being studied to reduce tissue damage, restore liver function, and reduce the high mortality rate on waiting lists [30]. For this purpose, stem cells are tested in clinical studies (Table 1).

Stem cells such as ESCs, HSCs, iPSCs, and MSCs can differentiate into hepatocyte-like cells, secreting soluble factors useful in treating liver diseases [31]. However, MSCs are most commonly used in clinical trials to restore liver injury and function (Table 1). These cells act in the liver environment, repairing the hepatic tissue and exerting anti-inflammatory, anti-fibrotic, anti-oxidative, and anti-apoptotic effects in vivo. Moreover, they improve liver function according to decreased prothrombin time and serum ammonia levels [32]. The differentiation of MSCs into hepatocyte-like cells can be achieved after incubation with some cytokines and growth factors, such as hepatocyte growth factors (HGFs), fibroblast growth factors-2/4 (FGFs), epidermal growth factors (EGFs), oncostatin M, leukemia inhibitory factor, dexamethasone, insulin-transferrin-selenium, or nicotinamide [33]. Considering, for example, the cirrhosis of different etiologies, an MSCs-based treatment was protective and had an anti-fibrotic effect [34] in addition to the hepatogenic differentiation and immunomodulatory function of MSC-derived microvesicles and soluble factors [35].

In the clinical trials shown in Table 1, MSCs were mainly harvested from bone marrow, the umbilical cord, and fat tissue. Bone marrow-derived MSCs (BM-MSCs) can promote hepatocyte regeneration and reduce liver stress and inflammation in vivo while playing an essential role in human patients’ liver mass regeneration and function [36]. The umbilical cord is also one of the most used anatomical sites to obtain MSCs for clinical trials due to their relatively easy access, abundance, and lack of ethical issues [37]. A pilot study used these cells to treat patients with biliary cirrhosis with no significant short-term or long-term complications. The patients’ life quality was improved after the transfusion, with fatigue and pruritus reduction, which are the most common complaints from those who suffer from this condition [38]. In addition, when used to treat liver cirrhosis and failure due to hepatitis B virus infection, the cells improved liver damage and function according to decreased alanine aminotransferase, glutamic-oxaloacetic transaminase, and total bilirubin levels [37].

IPSCs are also potentially good candidates for treating end-stage liver diseases. They can differentiate into functional hepatocytes in vitro, although this process is not easy and requires multiple steps. Studies in vivo showed that iPSCs could provide liver regeneration and secretion of liver proteins [36]. However, these cells can be tumorigenic and were not tested in clinical trials, requiring more in vitro and in vivo studies to determine their safety and efficacy in humans [39].

One of the major concerns and a key factor for cell-based therapies is the administration route, as the cells must reach the liver parenchyma [40]. Some administration routes were reported, such as infusion through the peripheral vein, which is the most common, the hepatic artery, the portal vein, intrasplenic, intrahepatic, and intraperitoneal injection. When MSCs are administered through the peripheral vein, there is an enrichment of injected cells in the liver parenchyma, especially in the case of chronic injuries. However, a more limited MSC engraftment is observed in the case of acute damage [33]. Regarding the injection of stem cells into the liver parenchyma, there is a higher risk of tissue damage, inflammatory infiltration, and rejection. Although the peripheral vein is apparently the safest route option, some studies showed that most cells injected are trapped in the lungs, reducing viable cell homing to the organ and differentiation in the liver, therefore decreasing treatment efficacy [4].

The number of cells injected and the correlation with the treatment’s efficacy must also be better investigated in clinical trials. Some studies suggest that the patients might not benefit from a unique high dose of cells. Therefore, splitting the total amount of cells to be injected in an extended period is recommended [37]. Compared to a single dose, repeated cell infusions induced prolonged clinical results and improved liver function [4]. These and other issues must be confirmed or defined with more randomized, controlled clinical trials and larger sample sizes [39]. Moreover, other technological innovations, such as nanotechnology, might help improve the efficacy and safety of stem cell-based therapy (Table 1). Clinical studies are found on www.clinicaltrials.gov (accessed on 15 March 2023) using the keywords “liver diseases” and “stem cells” simultaneously.

## 4. Nanotechnology

Nanotechnology is technological engineering at the nanoscale. The prefix “nano” is derived from the Greek word “nanos”, which means dwarf or something tiny and depicts one thousand billionths of a meter (10^−9^ m). Nanotechnology refers to structures, devices, and systems with novel properties and functions due to the arrangement of their atoms on the of nanoscales, i.e, at least one dimension from 1–100 nanometers (nm) [41] and is the size scale where quantum effects can determine the behavior and properties of particles. At the nanoscale, properties such as fluorescence, electrical conductivity, melting point, magnetic permeability, and chemical reactivity can change according to size. The same material can be better at conducting heat or electricity, become more chemically reactive, reflect light better, or change color as its size or structure is altered [42,43]. Since the materials’ properties can change significantly at larger scales, studying nanomaterials is essential to predicting and adjusting their response, fine-tuning the particles’ size to a specific application.

The development of nanotechnology, driven by advances in science and technology, clearly creates new opportunities for advancing medical science and disease treatment in human health care. Currently, nanotechnologies contribute to almost every field of science, including physics, materials science, chemistry, biology, computer science, and engineering. Particularly in cancer treatment, nanotechnologies are being applied to human health with promising results [44,45].

In the medical area (nanomedicine), nanomaterials application yielded a new range of highly selective and specific applications designed to maximize therapeutic efficiency and reduce side effects [46]. Numerous advantages can be achieved with the development of nanotechnology’s new applications. In the delivery of drugs, for example, it can improve the delivery of poorly water-soluble drugs, favor drug delivery into specific cells or tissues, or drug transcytosis across tight epithelial and endothelial barriers. Moreover, nanotechnology can help the delivery of large macromolecular drugs to intracellular sites of action, co-delivery of two or more drugs in combination, allow for better visualization of drug distribution by combining therapeutic agents with imaging probes, and allow for real-time reading of the in vivo efficacy of a therapeutic agent [47,48]. In the case of photothermal and photodynamic therapy, nanomedicine can provide better photosensitizers with higher photothermal conversion efficiency (PTCE) [47,48,49] and the capacity to generate photo-induced reactive oxygen species (ROS) [50,51,52]. Nanostructures can also be developed for bio-imaging technologies in the form of contrast enhancement nanoagents and nanoprobes. For example, nanostructures with broad absorption and narrow emission spectra are essential for fluorescence imaging [49,50], and magnetic nanomaterials can be used as contrast agents for magnetic resonance imaging (MRI) [51]. Moreover, nanomaterials with high PTCE can be employed for photothermal and photoacoustic imaging [52,53,54].

Numerous nanomaterials were studied in nanomedicine for various technological and biological applications. These nanomaterials can be organic or inorganic, composed of liposomes, dendrimers, polymeric nanoparticles (NPs) and micelles, graphene, carbon nanotubes, metal NPs, and quantum dots (QDs), for example, and all have advantages and disadvantages (Figure 1). Organic nanomaterials provide great flexibility in combining multiple functionalities and are applicable in regenerating bone, cartilage, skin, or dental tissues, for example [55]. However, this flexibility is combined with drawbacks, including intrinsic design complexity, high manufacturing cost, and structural instability. Inorganic nanomaterials are often intrinsically robust with relatively low manufacturing costs, but their limited design flexibility and functionality present challenges that remain to be fully overcome [54,56].

Cytotoxicity, rapid clearance from blood, and limited capacity to overcome multiple physiological barriers are critical issues for the clinical translation of nanomaterials. Nanoparticles can lead to toxic manifestations, resulting in allergy, fibrosis, and organ failure, in addition to hematological, neural, hepatic, splenic, nephron, and pulmonary toxicity [57,58,59,60]. Particle size and surface area also play significant roles in the interaction of materials with biological systems, in addition to particle shape and ratio, surface charge, aggregation capacity, surface coating and roughness, and solvents/media solubility [61] (Figure 1). These characteristics determine how the biological systems respond to nanomaterials and how they are distributed and eliminated [61]. In general, nanoparticles’ size-dependent toxicity can be attributed to their ability to enter biological systems [62] and modify tissue macromolecular structures [63], thereby interfering with critical bodily functions.

Many inorganic and even organic nanostructures exhibit poor biocompatibility and should be coated with biocompatible materials for biomedical applications [63]. After biological application, the nanostructures may not degrade in vivo or be eliminated by renal excretion, for example, possibly accumulating in particular organs and causing unwanted side effects. With new nanomaterials-based products frequently being introduced, gathering more information about physicochemical properties and their influence on material toxicity is necessary. Therefore, biocompatible nanomaterials with multiple functionalities are in great demand and represent a significant advance in several areas of medicine.

## 5. Nanotechnology and TERM (Tissue Engineering and Regenerative Medicine): A Rising Treatment Field

Technological progress and scientific advance played a significant role in the development of modern medicine, resulting in the emergence of new therapeutic approaches. These include cell therapy, a therapeutic approach still under development that proposes supplying viable cells to help the homeostatic reestablishment of injured tissues [64]. However, some aspects must be considered: First, the efficacy and safety of administering these cells. Then, a careful analysis of the administered cell’s functionality, ensuring their localization in the tissue of interest and lower chance of rejection by the recipient’s immune response [65]. With the advance of nanotechnology, large areas of chemistry, physics, engineering, biology, and materials science evolved significantly and now offer alternatives that mitigate these and other challenges [66].

As mentioned, nanoparticles have great versatility regarding their application [67,68] and composition, which includes carbon, magnetic nanomaterials, zeolites, and clay [69,70]. Amongst the challenges yet to be overcome in medical application is the nanoparticles’ capacity to traverse biological barriers such as endothelial surfaces, nanoparticle–protein interaction, tissue non-facilitated diffusion, phagocytic sequestration, and renal clearance. One strategy is using inflammatory cells as carriers, such as lymphocytes, monocytes/macrophages, dendritic cells, and neutrophils [71]. Leukocytes adhere firmly to the inflamed endothelium for transmigration due to interactions based on selectins and integrins. Then, they interact with stromal cells and ECM components while migrating in response to chemotactic stimuli. These intrinsic physiological characteristics make autologous leukocytes well-suited for carrying nanoparticles in health treatments. Another strategy is to prepare nanoparticles coated with leukocyte-derived plasma membranes, providing the nanoparticle with appropriate ligands to traverse barriers and reach the anatomical site that requires treatment [72] (Figure 1). For example, it was observed in vivo that administered leukocyte-based mimetic nanoparticles accumulated in inflamed tissues and remained in the site for about 8 h, facilitating tissue regeneration [73]. Consequently, the formulation of cellular-based drug-carrying nanoparticles using the cells of the immune system enables the targeted delivery of the pharmacological agent directly into the target tissue. This is possible because these cells are naturally recruited to inflammatory tissues, and a cell-based nanoplatform holds promise due to cellular biocompatibility [74]. However, some points need to be considered: First, the method of inserting the nanocarrier into the cells and the efficiency rate of nanoparticle engulfment. Considering the clinical management, it is advantageous to use autologous cells and a favorable chemotactic gradient, ensuring concentration of the drug dose [75].

Another technological innovation proposed to improve nanoparticle efficiency is adding photosensitive molecules [76] for internalization by specific cells. In photodynamic therapy, molecules activated by light (photosensitive), such as lasers or LEDs, are used to kill targeted cells [77]. This study observed a higher accumulation of nanospheres in the tumor tissue and slower blood clearance [78]. Mucoadhesive nanoparticles, such as those affecting the central nervous system, were also used for treating brain disease. However, the blood–brain barrier’s low permeability is likely to reduce effectiveness [79]. Additional conjugation of potentially therapeutic molecules, such as antibodies, peptides, growth factors, and other ligands may increase therapeutic action (Figure 1). These strategies can improve local targeting, nanoparticle solubility, stability, availability, and reduce the unwanted distribution in other sites in the body [80,81]. An inherent advantage of ligand association is producing specific nanoparticles for certain diseases with in-depth knowledge of the clinical scenario [82].

An example of innovative components associated with nanoparticles was recently published and described the treatment of experimental liver cancer using the delivery of RNA interference (RNAi). Extracellular vesicles isolated from milk were coated with RNA aptamers capable of binding to the epithelial cell adhesion molecule (EpCAM) [83], a marker of liver tumor cells, such as liver cancer stem cells (LCSC). The structure was also loaded with small RNAi to block the β-catenin pathway, an essential pathway in stem cell growth. In in vivo experiments using athymic nude/nude mice carrying a liver cancer xenograft, complexed nanoparticles accumulated more in tumors with higher EpCAM expression. Accordingly, in these mice, the tumor regressed faster [83].

In another experiment, MSCs loaded with gold nanoparticles conjugated with superparamagnetic iron oxide (SPIO) were used to treat an induced liver injury in athymic nude/nude mice. One factor the authors considered is that when irradiated with infrared light, the nanoparticles induced thermal ablation by raising the local temperature and cell death [84]. Then, magnetic resonance analysis showed the hepatic tissue’s darkening after 3 h of administration, which persisted for 24 h. These findings illustrate the numerous structural modifications, components, and strategies that can be combined into nanoparticles to treat multiple disorders.

Graphene oxide-based nanomaterials are also good candidates for liver and other organ treatments in nanomedicine. Graphene oxide is a graphene derivative with a two-dimensional honeycomb structure; it is a biocompatible, non-toxic, and water-dispersible nanomaterial that efficiently absorbs proteins and other small biological components [85]. However, it is hard to separate these structures by conventional laboratory methods, so it can be associated with Fe_3_O_4_ superparamagnetic nanoparticles to be easily separated by a magnetic field (Figure 2). Using this approach, a magnetic graphene oxide nanostructure was associated with a conditioned medium (CM) of ESC-derived MSC (ESC-MSC-CM) culture and injected into rats with acute liver failure (ALF). It was observed that the treatment reduced liver necrosis and inflammation and increased the vascular endothelial growth factor and matrix metalloproteinase-9. The treatment using ESC-MSC-CM was more effective than the injection of CM only [85,86]. In another study, a nanosheet of silica magnetic Fe_3_O_4_-graphene oxide (SMGO) associated with MSCs’ CM (SMGO-CM) was prepared and administered to rats with ALF. The group that received SMGO-CM showed reduced liver damage and inflammation, with increased regeneration and survival [87]. Therefore, the use of nanoparticles functionalized with biomolecules and improved half-life in blood circulation increases targeting specificity and helps the recovery of damaged tissues [88]. For this purpose, different structures can be incorporated into nanoparticles as surfactants [89], dendrimers [90] polymers [91], and biomolecules [89] (Figure 2). However, it should be noted that the wide application of functionalized nanoparticles needs to satisfy certain criteria, such as low cytotoxicity, low interaction with plasma proteins, low recognition by the mononuclear phagocytic system, good colloidal stability, and release of the therapeutic agent at the ideal dose and in the predetermined target tissue [92].

## 6. Mechanisms for Nanoparticles Tissue Targeting

It is possible to analyze if administered nanoparticles reached the intended target tissue. In the case of tumors, the accumulation may occur due to a vascular phenomenon named enhanced permeability and retention effect (EPR) [93]. As most solid tumors receive a greater blood supply due to neovascularization, the nanostructures, macromolecules, and nutrients, tend to be better retained [94]. Moreover, some requisites should be observed, considering that intravenous injection may lead to blood clot formation and immune activation. The aggregation capacity, half-life in blood, size less than 400 nm, hydrophobicity, and surface charge should be determined to avoid plasma proteins adsorption and other potential side effects [95]. This approach’s efficiency also depends on the stage and type of tumor and local vascularization [96].

The conjugation of nanoparticles with selected molecules can increase therapeutic results; these include antibodies, bioactive peptides, growth factors, and several others. The nanostructures can also be combined with components that target the particles to specific tissues, improve the solubility and availability, minimize surface energy, and more [80,81]. An inherent advantage of associating multiple ligands and therapeutic components is producing specific nanoparticles for certain diseases [82].

Another mechanism for nanoparticle tissue targeting is based on magnetic forces, and this approach can also be used to reduce drug doses and side effects [97]. The main magnetic materials in biomedicine are metal oxides such as magnetite (Fe_3_O_4_) and ferrites, including CoFe_2_O_4_ NiFe_2_O_4_. Since iron is vulnerable to corrosion and rust in water, a non-porous coating is essential, and iron alloys such as FePt and FeAu are frequently used [98]. It was published that synthesized magnetite nanoparticles of about 80 nm are good vehicles for biomedical applications. Their therapeutic potential was tested in vitro using the hepatocellular carcinoma cell line HepG2 [99] at different concentrations, and they induced cellular proliferation and production of ROS in a dose- and time-dependent matter. Although magnetite nanoparticles have no therapeutic potential themselves, they can be subjected to a magnetic field for biological effects. When a magnetic force was applied at twenty-four and seventy-two hours, the mitochondrial activity increased, a phenomenon not observed without the magnetic field. According to the authors, this can be explained by the ability of cancer cells to regulate the gene expression of proteins involved in iron absorption and the regulation of tumor signaling pathways such as a hypoxia-inducible factor (HIF) [99].

It was evaluated by MRI whether injected primary hepatocytes loaded with SPIO nanoparticles, combined with protamine sulfate, would be retained in the liver. Before the injection, it was observed that cellular absorption of iron was around 80%, showing good cell viability and normal secretion of lactate dehydrogenase (LDH), albumin, and urea for up to fourteen days in culture. Labeled hepatocytes were then used in rats for intrasplenic transplantation after induced ALF. The authors identified a significant iron increase in periportal areas, endothelial cells, and Kupffer cells, suggesting the translocation of hepatocytes to the liver and their phagocytosis. Therefore, although the cellular loading with magnetic nanoparticles was optimized, cell viability is yet an issue, at least for long-term analysis [100].

SPIO magnetic nanoparticles are increasingly being used in biotechnology and biomedicine, mainly for being a contrast agent in MRI. Moreover, due to its biocompatibility, when SPIO interacts with cells, it provides a magnetic field with oscillations, consequently inducing a phase shift of protons, which facilitates their detection by MRI [101]. Despite its biocompatibility, SPIO has an aggregatory nature, requiring modifications on its surface to minimize, for example, interactions with plasma proteins and elimination by resident tissue macrophages, which can lead to inflammatory processes, thrombosis, and anaphylaxis [102]. In the literature, it was reported that coating SPIO with poly(ethylene glycol) gives the nanoparticle a greater hydrophilicity and minimizes nonspecific interactions with other biomolecules. However, factors such as the hydrodynamic size of the coated nanoparticles and the nature of the crosslinking agent will directly affect the half-life in blood and iron absorption by macrophages [103]. Another class of polymers used was monosaccharides, which aim to provide the nanoparticle with nonspecific adsorption prevention, specific targeting, and cell internalization. This is possible because carbohydrates attached to the surface can connect to cell membrane receptors and direct nanoparticles to subcellular compartments [104]. However, many studies must be directed to understand the interaction between SPIO and the extracellular and intracellular environments.

As previously mentioned, the aggregation capacity is a factor to be considered, because when nanoparticles are administered in vivo, several proteins can be adsorbed in the particle’s surface and confer a new biological identity. In short, in in vivo systems, low-affinity proteins bind weakly to the nanomaterial structure, being a marker for other serum proteins to bind to the structure and leading to particle degradation by the mononuclear phagocytic system [105]. The binding capacity of serum proteins will directly depend on the size, morphology, surface characteristics (curvatures), and hydrophobicity of the nanomaterial [106]. Given the administration in the bloodstream, the plasma proteins will be adsorbed in the structure of the SPIOs, increasing their hydrodynamic size, a biological mechanism known as opsonization that facilitates the interaction and binding of the mononuclear phagocytic system. This leads to the degradation of the nanoparticles present in the bloodstream. Moreover, nanoparticles tend to accumulate in the liver and spleen, where macrophages phagocytose xenobiotic particles and eliminate them from the body [107].

Regarding the mechanisms that lead to magnetic nanoparticle degradation, it is known that insoluble ferric ions (Fe^3+^) bind to free transferrin in the blood. Then, this protein transports the ion and mediates the binding to the cell membrane transferrin receptor (TFRC). Intracellularly, Fe^3+^ is converted to a ferrous ion (Fe^2+^) under acidic conditions by the enzyme metalloreductase (STEAP3). Once converted, Fe^2+^ can be stored in ferritin or oxidized and exported to the extracellular environment by ferroportin (FPN) [108]. With an excess of Fe2+ in the intracellular medium, it can be used as a catalyst to convert hydrogen peroxide (H_2_O_2_) into hydroxyl radicals (OH), a redox process known as the Fenton reaction [109]. In turn, free OH can interact with polyunsaturated fatty acids of the lipid membrane and induce lipid peroxidation [110]. Under normal conditions, formed lipid hydroperoxides can be neutralized by the enzyme glutathione peroxidase 4 (GPX4), a selenoprotein that uses free glutathione (GSH) in the cell to reduce lipid hydroperoxides into their respective alcohols. Thus, this mechanism protects the cells against oxidative stress [111]. However, under stress conditions, excess of intracellular free iron can increase phospholipid oxidation, leading to the degradation of plasma membrane phospholipids and consequently favoring oxidative cell death, a process known as ferroptosis. Some factors contribute to ferroptosis-mediated cell death, such as the inhibition of the cystine-glutamate (Xc-), an anti transport system by erastin, directly affecting intracellular GSH replacement [112]. Another factor is the binding of the RSL3 molecule to GPX4, inhibiting its catalytic activity. The enzyme inactivation causes the accumulation of lipid peroxides, resulting in increased ROS [113]. Thus, ferroptosis induction by free iron excess is a major concern in applying magnetic nanoparticles such as magnetite and maghemite (Fe_2_O_4_).

It was seen that magnetic nanoparticles loaded with sulfasalazine (SAS) were “camouflaged” in the platelet membrane (PLT), forming a Fe_3_O_4_-SAS@PLT complex with a hydrodynamic size of 268 nm. When used in in vitro experiments, an increase in the levels of ROS and lipid peroxides, depletion of GSH and the XcT system was observed, factors that indicated that the ferroptosis pathway was activated. In in vivo models, the authors observed that GPX4 expression in mice was reduced after treatment with Fe_3_O_4_-SAS@PLT, indicating that ferroptosis may be involved in controlling metastatic tumors. The strategy, in this case, was to combine ferroptosis with immunotherapy to eliminate metastatic cells and increase treated animals’ survival [114]. A recent study produced gelatin microspheres loaded with magnetic nanoparticles and the drug adriamycin (ADM) to treat hepatocellular carcinoma in a combined therapy, using radiofrequency hyperthermia and chemotherapy. When analyzing the participation of ferroptosis, the data indicate that in the presence of ferroptosis inhibitors with ADM/Fe_3_O_4_-MS, the viability was greater when compared to the group without inhibitors. In in vivo models, a combined therapy also reduced liver tumor mass, demonstrating antitumor efficiency compared to untreated groups [115].

Recently, a nanostructured platform sensitive to pH and redox was used for tumor therapy. This platform was composed of Fe_3_O_4_ nanoparticles encapsulated in mesoporous structures of organophilic silica probe (MONs). They were then loaded with the drug sorafenib, manganese dioxide, and modified with hyaluronic acid (HA) and glucose oxidase (GOD) to facilitate cellular internalization. The study showed that this nanoplatform had a hydrodynamic diameter of 150 nm. The drug release rate evaluated was low at neutral pH (8.3–10.4%) and high within lysosomes, reaching 72.3%. These results indicate a low probability of the drug being released into the bloodstream. In addition, the possibility of hemolysis and elimination of nanoparticles in the blood by disintegration was low, which makes them safer for human administration. The cytotoxicity of the nanostructure evaluated in vitro, using human lung adenocarcinoma cells (A549), proved to be low and with high intracellular uptake. The antitumor therapy analysis performed on BALB/c-nu mice bearing tumors indicated that the nanosystem directed by the magnetic field significantly reduced the tumor mass compared to the control group. The histological analysis indicated ferroptosis, which potentially caused the regression of the tumor tissue [116].

Magnetic nanoparticles were used not only to induce cell death by ferroptosis, but also to detect tumor cells [117]. This is particularly relevant because, currently, the most common method used to detect tumor cells is based on biopsies, invasive procedures for histopathological analysis [118]. By using magnetic nanoparticles, it is possible to detect tumor cells less invasively by MRI. Thus, nanoparticle formulations made of Fe_3_O_4_ were modified with dimercaptosuccinic (DMSA) and coated with gold, forming Fe_3_O_4_-Au with a particle size of 28 nm. When tested on gastric carcinoma cells, the nanoparticles demonstrated low cytotoxicity. The attenuation capacity of the analyzed X-rays showed an increase in intensity depending on the concentration of nanoparticles. Another factor that contributed to high radiographic attenuation was the ultrafine structure of the nanoparticles, enabling increased contrast by computed tomography (CT). When tested in an in vivo model, these nanoparticles predominantly accumulated in the liver after 45 min of injection, which was demonstrated by the darkening of the liver tissue when analyzed by CT and confirmed by MRI. These results were validated by applying the methodology to mice affected by non-alcoholic fatty liver disease. In this case, the CT image better showed the accumulation of nanoparticles in the entire liver tissue, different from what was observed in the untreated group, indicating a high uptake of magnetic nanoparticles.

Dual modality contrast CT/MRI performed to detect orthotopic liver cancer was facilitated after nanoparticles administration, showing the border of liver lesions. These lesions were confirmed by pathological analysis that showed nuclear pleomorphism and hypercholemasia, multinucleation of hepatocytes, and infiltration of tumor cells. Although the authors were concerned with in vivo cytotoxicity, histological studies confirmed nanoparticles’ presence in the liver and spleen after 24 h of treatment, with low infiltration of immune cells. In the long term, tissues collected for analysis did not show acute injuries or post-administration chronic inflammation [119]. Another study aimed at treating liver cancer using formulated magnetic nanoparticles loaded with doxorubicin (DOX) and functionalized with poly(ethylene oxide)-trimellitic chloride-folate anhydride (PEO-TMA-FA). This antitumor nanoplatform treatment was performed in rabbits with xenografts of VX2 liver tumors, a metastatic tumor, and a tumor reduction was morphologically and macroscopically observed. Moreover, fluorescence analyses indicated that this high potency inhibited tumor proliferation and angiogenesis [120].

In conclusion, most studies using magnetic nanoparticles direct efforts for application in oncology due to their extrinsic characteristics, such as magnetic field targeting [121] and elimination of tumor cells by hyperthermic therapy [122,123] or by ferroptosis [124]. However, for the regeneration of injured tissues, one possibility would be to produce magnetic microcapsules or nanocapsules loaded with molecules that could help in the recovery of damaged tissue, such as: targeting EGFs and HGFs to be released locally and help with remodelling the hepatic matrix [25,125]. Another possibility would be the use of liver cells or stem cells as nanoparticle carriers. However, a delicate point to be analyzed would be the ideal concentration of magnetic nanoparticles in the polymeric matrix that would respond to the magnetic field, since the concentration of ferrous ions could lead to cell death or even impair tissue recovery. Thus, there are countless possibilities to be studied and many doors to be opened in this field.

## 7. Applications of Stem Cells in Nanomedicine

Given promising results, a new area of study is emerging in regenerative medicine and tissue engineering, known as “stem cell nanotechnology”. In this field, nanotechnology improves the efficiency of stem cells’ therapeutic action, promoting healing and tissue repair [126]. A study published in 2016 used bioactive glasses and polymers as scaffold nanomaterials to deliver stem cells in the bone tissue for regeneration. This strategy was used because bioactive glasses have high osteoconductivity and biocompatibility and can interconnect with bone tissue. Moreover, gelatin was used as a porous material since it is commonly used to structure the ECM. The study aimed to compare the effect of different stem cells (BM-MSCs, UC-MSCs, and AD-MSCs) on bone regeneration using nanocomposite scaffolds made of bioactive glass and gelatin. The study showed that the scaffolds used were not cytotoxic and suitable for cell development and expansion. To analyze the regenerative capacity of this bioconstruct, an injury was made in the calvaria area of the skull in male Wistar rats after four to twelve weeks of implantation. Bone regeneration depended on the implantation time, and tissue healing was better in rats that received the implant. They also observed that BM-MSCs cells had more potential for differentiation and bone consolidation when compared to the other cell types. On the other hand, UC-MSCs had a greater capacity for angiogenesis [127].

Another study used the synthetic polymer polycaprolactone (PCL) with collagen (Figure 1). The authors employed the electrospinning technique, which produces aligned nanofibers with a high surface area, high porosity, and a functional surface that provides mechanical stability, structural orientation, and cell adhesion. In this study, they used stem cells extracted from the bone marrow, and the results indicate that the cells had a high proliferation rate in this scaffold. Moreover, they presented neuronal-like morphology after twenty-eight days in culture in the scaffold and with stimulation to differentiate into neuronal cells. Considering that this structure was adequate to support cell proliferation and differentiation, the authors suggest that this biocompatible nanofibrous support can be used in neuro transplantation [128].

Recently, a study aiming at bone regeneration used porous silica coated with magnetic nanoparticles to stimulate MSCs differentiation. It supported bone differentiation of MSCs in vitro with increased bone matrix mineralization, while in vivo experiments using rats showed bone regeneration after nanoparticle administration [129]. In addition, nanoparticles were used to target and deliver non-phagocyte stem cells utilizing the magnetofection. This transfection technique uses magnetic fields to concentrate particles containing vectors in targeted cells in the body. Nanoparticles coated with polyethylene glycol (PEG) were used to direct human MSCs with PEG magnetic-specific features showing no toxicity or interference in functional cell characteristics. However, more studies are necessary to improve this technique [130]. Ferumoxytol, a magnetic nanoparticle approved by the United States of American Food and Drug Administration (FDA-USA), was used in infarcted rats to target and direct rat cardiosphere-derived stem cells (rCDCs). This approach improved damaged cardiac tissue recovery, indicating a long-term graft possibility [131].

MRI showed that rat MSCs labeled with Fe_3_O_4_ migrated to the hepatic fibrotic tissue after one hour in a liver fibrosis model [132]. Another study reported that stem cells labeled with a magnetic nanomaterial were retained in the liver after intrahepatic administration. However, both studies reported the loss of MRI signal after seven days, indicating that the cells died, migrated to other tissues, or were endocytosed by Kupffer cells [133]. Another therapeutic possibility is the application of stem cell membranes to coat nanoparticles. This study used membrane-coated nanoparticles containing the DOX drug for colon cancer treatment. The nanostructures had preserved membrane proteins and good stability over fourteen days in vitro, and the diameter was compatible with renal excretion. The drug release kinetic was evaluated at different pHs, and it was observed that 60% of the drug was fully released within 36 h at pH 5, unlike pH 7.5, when only 20% of the drug was released. Apparently, the membrane helped to control the drug release, and the cytotoxicity of the nanostructure-associated drug was lower than the drug diluted in a medium. Finally, the in vivo experiments using mice showed a significant reduction in tumor growth [134]. Multiple advantages are associated with this technology, such as phenotype conservation, ease of in vitro expansion to obtain stem cell membranes, and a high potential for intrinsic migration to regions of high inflammation, which facilitates specific drug delivery [135]. Moreover, it has low immunogenicity, increased time for blood clearance due to its “camouflage”, low neurotoxicity, and the capacity to interact with receptors and molecules on other cells in the tissues [136].

Another promising possibility is using stem cells loaded with magnetic nanoparticles to treat acute or chronic liver diseases, such as cirrhosis, ALF, hepatitis, alcoholic or non-alcoholic liver diseases, and hereditary liver diseases. The central idea would be to administer modified stem cells and subject the individual to a magnetic field to concentrate the loaded cells in a specific anatomic location (Figure 3).

Numerous possibilities are yet to be tested and developed in regenerative and nanomedicine using multiple combinations and technological advances likely to affect clinical management profoundly.

## 8. Conclusions and Future Perspective

Although there are numerous challenges to be overcome concerning the most appropriate type of stem cell, ethical issues, infusion routes, number of cells, protocols, specific functions, and toxicity of magnetic nanoparticles aimed at cell transplantation, intensive scientific research is being conducted in this area. In the future, these approaches can help to support liver transplantation, either increasing life expectancy until the surgical procedure or helping the recovery of liver damage. In this context, nanomedicine is gaining prominence in the scientific community due to the formulations of biomimetic and functionalized nanomaterials that enable the delivery of genes, drugs, and cells, in addition to being used as a support for cell growth, mimicking the ECM. These approaches can be combined with other methodologies, such as photodynamics and sonodynamics, enabling a higher therapeutic potential. However, it is crucial to consider the limits of using these nanosystems, such as in vivo toxicity, retention site, and possible adverse effects. In addition, it is necessary to assess the impact of these approaches on the public and private health systems and accessibility for the population. The production of functional nanomaterials for therapy requires ultrafine resolution methodologies and experimental studies in animal models to confirm human safety. These are essential steps that require efforts by the scientific community to produce effective nanoplatforms for regenerative or oncological therapy.

## Figures and Tables

**Figure 1 ijms-24-09293-f001:**
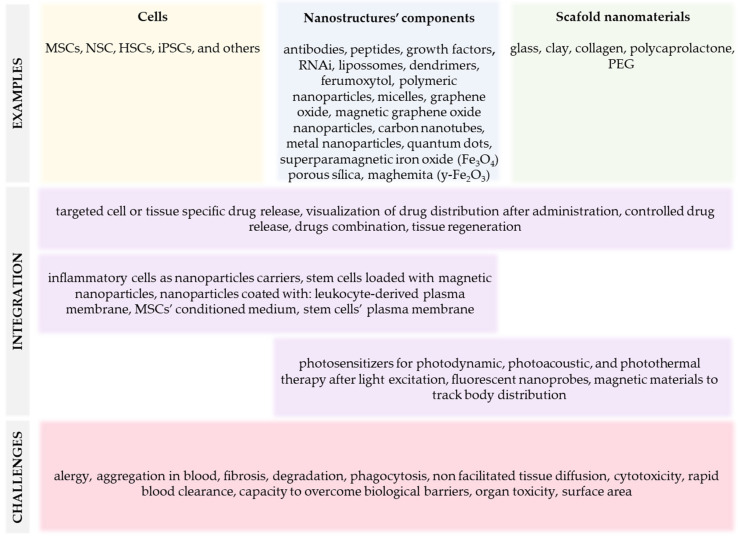
Summary of nanomaterials-based alternative therapies to treat liver diseases.

**Figure 2 ijms-24-09293-f002:**
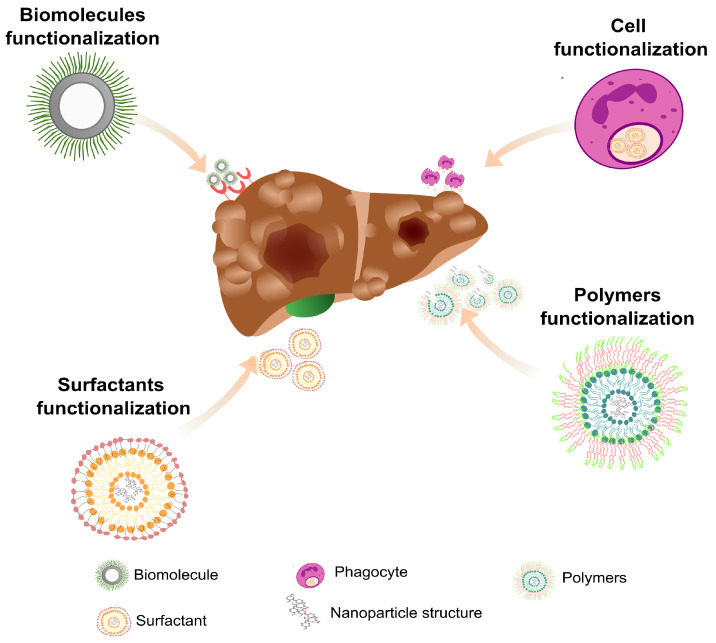
Functionalization of nanoparticles for the treatment of liver damage. The surface of most nanomaterials can be modified, allowing for the insertion of molecules that can act directly in the therapeutic approach or help in the treatment itself. Surface functionalization minimizes nanoparticle clearance, decreases the likelihood of therapeutic agent release in uninjured sites, and makes the nanoparticle more biocompatible.

**Figure 3 ijms-24-09293-f003:**
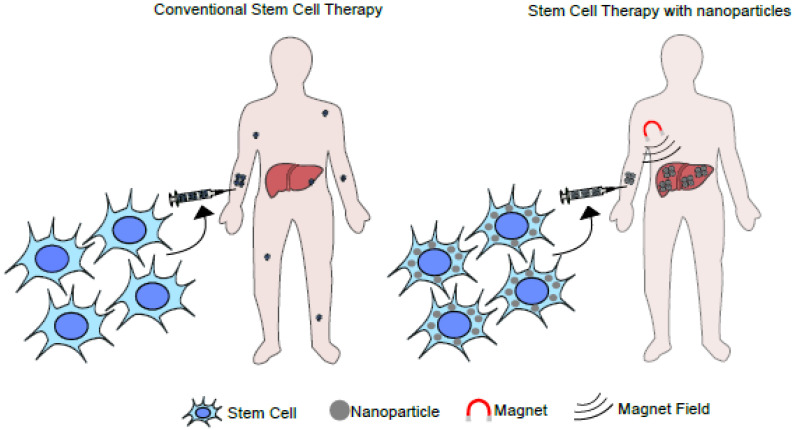
Illustration of a liver-targeted stem cell-based therapy using a magnetic field. The diagram illustrates the application of an external magnetic field over systemically applied magnetic nanoparticles. Under this setting, the nanoparticles can be enriched in a particular anatomic area or organ.

**Table 1 ijms-24-09293-t001:** Clinical studies identified on clinicaltrials.gov that have currently used stem cells in liver disease.

Identifier at ClinicalTrials.gov	Liver Condition	Stem Cell Type *	Study Phase	Enrolment/ Estimated Enrolment	Status	Administration Route and Cell Dose
NCT03109236	Cirrhosis	Autologous EPC CD133^+^ from BM	Phase 3	66 participants	Recruiting	5–10 × 10^6^ CD133 cells through the transhepatic route into the portal venous circulation.
NCT05331872	Cirrhosis	UC-MSCs	Phase 1	20 participants	Recruiting	Route not informed. Cell dosage not informed.
NCT05227846	Cirrhosis	UC-MSCs	Phase 1	9 participants	Recruiting	Cell dosage not informed.
NCT03945487	Cirrhosis	UC-MSCs	Phase 2	200 participants	Recruiting	Intravenous administration of 1.0 × 10^6^ cell/kg three times at three-week intervals.
NCT05121870	Cirrhosis	UC-MSCs	Phase 2	240 participants	Recruiting	Intravenous administration of three doses (6.0 × 10^7^ cells per event) at weeks 0, 4, and 8.
NCT03626090	Cirrhosis	Autologous BM-MSCs	Phase 1/2	20 participants	Recruiting	A single dose of 0.5 to 1 × 10^6^ cells/kg via peripheral venous access.
NCT03254758	Cirrhosis	AD-MSCs	Phase 1/2	27 participants	Recruiting	Intravenous infusion; for phase 1, the cell dose escalated from low to mid and high; for phase 2, the recommended amount of cells was administered once a week for four weeks in the same route and time as in phase 1. Cell dosage not informed.
NCT05155657	Alcoholic cirrhosis	UC-MSCs	Phase 1	36 participants	Recruiting	0.5 × 10^6^ cells/kg, 1.0 × 10^6^ cells/kg, or 2.0 × 10^6^ cells/kg via intravenous infusion.
NCT04689152	Alcoholic cirrhosis	Autologous BM MSC	Phase 3	200 participants	Recruiting	7 × 10^7^ cells via the hepatic artery.
NCT03826433	Cirrhosis due to hepatitis B	UC-MSCs	Phase 1	20 participants	Recruiting	6 × 10^7^ cells via peripheral intravenous injection.
NCT05507762	Cirrhosis due to hepatitis B (compensation stage)	UC-MSCs	Phase 1/2	20 participants	Recruiting	1 × 10^6^/kg/time per injection via intravenous infusion in the elbow.
NCT05106972	Cirrhosis due to hepatitis B	UC-MSCs	NT	30 participants	Recruiting	1 × 10^8^ cells/dose via intravenous infusion.
NCT00655707	Liver disease	Autologous expanded CD34^+^ HCSs	Phase 1/2	5 participants	Completed	1 × 10^9^,1 × 10^10^, 2 × 10^10,^ or 5 × 10^10^ cells via either the hepatic artery or the portal vein.
NCT00420134	Liver failure /cirrhosis	Autologous MSCs	Phase 1 Phase 2	30 participants	Completed	The cells were administered via the portal vein. Cell dosage not informed.
NCT00147043	Cirrhosis	Autologous adult stem cells	Not Applicable	5 participants	Completed	The cells were administered via the hepatic artery or portal vein. Cell dosage not informed.
NCT04243681	Cirrhosis	Autologous CD34^+^ HSCs and MSCs	Phase 4	5 participants	Completed	The cells were administered via the hepatic artery. Cell dosage not informed.
NCT00713934	Cirrhosis	Autologous BM-MNCs and enriched CD133^+^ HSCs	Phase 1/2	7 participants	Completed	The cells were administered via the portal vein. Cell dosage not informed.
NCT02297867	Cirrhosis	ADSCs	Phase 1	6 participants	Completed	One milliliter of cell suspension via intrahepatic injection.
NCT03632148	Cirrhosis	MSCs	Not applicable	9 participants	Completed	Route not informed. Cell dosage not informed.
NCT01342250	Cirrhosis	UC-MSCs	Phase 1 Phase 2	20 participants	Completed	The cells were administered at low, medium, or high doses. Route not informed. Cell dosage not informed.
NCT01333228	Cirrhosis	Autologous BM-derived endothelial progenitor cells	Phase 1/2	14 participants	Completed	The cells were administered via the hepatic artery. Cell dosage not informed.
NCT01013194	Cirrhosis	Human fetal liver cell	Phase 1/2	25 participants	Completed	5 or 10 × 10^8^ cells via the splenic artery infusion
NCT01454336	Liver cirrhosis /fibrosis	Autologous MSCs	Phase 1	3 participants	Completed	The cells were administered via the portal vein. Cell dosage not informed.
NCT01220492	Cirrhosis	UC-MSCs	Phase 1/2	266 participants	Completed	The cells were administered once a week for four weeks at a dose of 0.5 × 10^6^ /kg body and intravenously for eight weeks.
NCT01120925	Cirrhosis	BM-MNCs and enriched CD133^+^ HSCs	Phase 1/2	30 participants	Completed	BM-MNC were administered at a dose of 2–3 × 10^9^ cells and CD133 at a dose of 5–15 × 10^6^ cells, both via the portal vein.
NCT03963921	NASH—non-alcoholic steatohepatitis	Liver-derived MSCs	Phase 1/2	23 participants	Completed	Route not informed. Cell dosage not informed.
NCT01591200	Alcoholic liver cirrhosis	Allogeneic BM-MSCs	Phase 2	40 participants	Completed	The cells were administered at a low, medium, or high dose via the hepatic artery. Cell dosage not informed.
NCT01875081	Alcoholic cirrhosis	Autologous BM-MSCs	Phase 2	72 participants	Completed	5 × 10^7^ cells via the hepatic artery.
NCT01378182	Wilson’s cirrhosis	Allogeneic BM-MSCs	Not applicable	10 participants	Completed	1 × 10^6^ cells/kg in total, with 1/2 of the dose in the peripheral vein and 1/2 of the dose in the right hepatic artery.
NCT01062750	Cirrhosis	Autologous adipose tissue-derived stromal cells	Not applicable	4 participants	Completed	The cells were administered via the hepatic artery. Cell dosage not informed.
NCT00956891	Liver failure due to hepatitis B	Autologous BM-MSCs	Not applicable	158 participants	Completed	The cells were administered via the hepatic artery. Cell dosage not informed.
NCT05517317	Liver cirrhosis due to biliary atresia	Autologous BM-MNCs	Phase 1	12 participants	Completed	The cells were administered via the hepatic artery. Cell dosage not informed.

* UC-MSCs: umbilical cord mesenchymal stem cells; AD-MSCs: adipose-derived mesenchymal stem cells; BM-MSC: bone marrow mesenchymal stem cells. EPCs: endothelial progenitor cells; BM: bone marrow; BM-MNC: bone marrow mononuclear; and HSCs: hematopoietic stem cells.

## Data Availability

Not applicable.

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
