# Peer review of "Magnetic Nanostructures and Stem Cells for Regenerative Medicine, Application in Liver Diseases"

_ijms, 2023, doi:10.3390/ijms24119293_

Round 1

Reviewer 1 Report

The content of the manuscript does not match the title because much of the article is about stem cells for liver regeneration while the title says it about nanomaterials for stem cell regneration. I recommend the manucript be rewritten so all the stem cell studies are deleted, and the remaining article focus only on nanomaterials. I also recommend the review of nanomaterials focus on limitations of published studies, i.e. a more critcal review be written, as that would be a better contribution to the literature.

overall the english is acceptable.

Author Response

Response: We appreciate your comment, and we added this view to the manuscript. We agree with the statement regarding the title and content of the manuscript. Therefore, we changed the title by adding the term "stem cells" and reduced the topics related to this subject, as highlighted in the text. However, we kept the main content about stem cells as we aim to also highlight the importance of using them for regenerative medicine. We do hope these alterations will meet the reviewer’s expectation.

Reviewer 2 Report

Authors have chosen an interesting topic for review. However, there is no clarity on what is the outcome of this review and what is focus for writing this review. Message to readers is not clear enough to understand.

1. Abstract has been redefined to keep the focus on the title chosen.

2. Authors started manuscript by explaining about the stem cells, bioengineering and regenerative medicine, which is not again direct relevant to chosen topic. Instead, authors should start the manuscript by highlighting the recent trends, scope and opportunities of nanotechnology for stem cell rejenuvation.

3. In the section 3, authors have generalized the nanotechnology applications in regenerative medicine. Instead they should focus on how " Nanostructures or Nanomaterials plays role in Liver regeneration (Hepatic cells) as well as importance of magnetic nanoparticles, to be specific.

4. In section 4, 5 & 6 similar corrections have to be made by authors. Authors have to keep the focus on Nanostructures or Nanomaterials applications in cytotoxic on hepatic cells, regenerative properties of magnetic nanomaterials on hepatic cell differentiation, regeneration and maintenance. Basically, highlighting on the title of the manuscript.

5. It is recommended to included clinical data and animal studies data of proving the impact of nanostructures for regenerating liver tissues. 

6. It is recommended to improvise the graphical representation.

7. It is recommended to cite only the recent published articles. (past 5-6 years only). 

Authors need to improvise the quality of English.

Author Response

Independent Review Report, Reviewer 2

“Authors have chosen an interesting topic for review. However, there is no clarity on what is the outcome of this review and what is focus for writing this review. Message to readers is not clear enough to understand.

  1. The abstract has been redefined to keep the focus on the title chosen.

Response: We do appreciate this comment. Our goal from the beginning was to emphasize the importance of targeting cells by using magnetic nanoparticles for the treatment of diseases, especially liver diseases. In this context, the use of stem cells is a promising approach highlighted in the literature. As noted by the first reviewer in the previous version, we believe that by not including the term "stem cells" in the title, the extensive description of this subject included in the text was confusing to the reader, although related to our main proposal. Therefore, we believe that we solved this issue by modifying the title and reducing some passages in the manuscript about stem cells in particular. We hope we achieved the expected equilibrium regarding all the topics approached, leading to a clearer text and a clear understanding of our focuses.

  1. Authors started manuscript by explaining about the stem cells, bioengineering and regenerative medicine, which is not again direct relevant to chosen topic. Instead, authors should start the manuscript by highlighting the recent trends, scope and opportunities of nanotechnology for stem cell rejenuvation.

Response:To meet this need, we have included a topic introducing the most relevant issues of our review proposal to clarify our objective.

  1. In the section 3, authors have generalized the nanotechnology applications in regenerative medicine. Instead they should focus on how "Nanostructures or Nanomaterials plays role in Liver regeneration (Hepatic cells) as well as importance of magnetic nanoparticles, to be specific.

Response: To better contextualize the subject described in topic 3 and follow the abovementioned suggestion, we have included new paragraphs in the text (all marked in red).

  1. In sections 4, 5 & 6, similar corrections have to be made by the authors. Authors have to keep the focus on Nanostructures or Nanomaterials applications in cytotoxic on hepatic cells, regenerative properties of magnetic nanomaterials on hepatic cell differentiation, regeneration and maintenance. Basically, highlighting on the title of the manuscript.

Response: Modifications were made as suggested and are highlighted in the manuscript using the "Word Tracker changes tool”.

  1. It is recommended to included clinical data and animal studies data of proving the impact of nanostructures for regenerating liver tissues.

Response: As the reviewer recommended, we included some examples of the suggested subject in the text.

  1. It is recommended to improvise the graphical representation.

Response: Modifications were made as suggested. For this, we included two new figures.

  1. It is recommended to cite only the recent published articles. (past 5-6 years only).

Response: With the new textual changes, we included more recent references related to the theme, as suggested. However, some references are related to original concepts, so we believe we should not remove them from the manuscript.

Reviewer 3 Report

The current work focuses on MAGNETIC NANOSTRUCTURES IN REGENERATIVE MEDICINE, APPLICATION IN LIVER DISEASES. The author’s some effort into the manuscript, but major issues should be addressed.

Abstract

-Magnetic nanostructures for bio application have been extensively studied. First, show the current review's novelty and importance and then the main outputs.

-The review is based on magnetic structure!! No word “magnetic” in the abstract!!

-First appearance abbreviation e.g. COVID-19 should have a full definition

-The toxicity of nanoparticles for the treatment is essential and should be clear. A subsection related to toxicity should be inserted to clear this point.

-Line 330, under this section “applications of stem cells in nanomedicine” more details are required to write with the most recent related references

-There is a felt lack of critical assessments by the authors. The authors did not mention the research gap between the previously reported articles and the present situation. Authors should incorporate their views in each subsection to mold the research in a new direction.

-Where Conclusion and future perspective!!

Extensive editing of English language required

Author Response

The current work focuses on MAGNETIC NANOSTRUCTURES IN REGENERATIVE MEDICINE, APPLICATION IN LIVER DISEASES. The author’s some effort into the manuscript, but major issues should be addressed.

-Magnetic nanostructures for bio application have been extensively studied. First, show the current review's novelty and importance and then the main outputs.

-The review is based on magnetic structure!! No word “magnetic” in the abstract!!

-First appearance abbreviation e.g., COVID-19 should have a full definition

-The toxicity of nanoparticles for the treatment is essential and should be clear. A subsection related to toxicity should be inserted to clear this point.

-Line 330, under this section “applications of stem cells in nanomedicine” more details are required to write with the most recent related references

-There is a felt lack of critical assessments by the authors. The authors did not mention the research gap between the previously reported articles and the present situation. Authors should incorporate their views in each subsection to mold the research in a new direction.

-Where Conclusion and future perspective!!”

Response: We appreciate the comments and have made significant changes to our manuscript to meet the needs suggested by the reviewer. All modifications included in the text are highlighted using the “Word Tracker Changes tool”. With this, we hope to have met the points mentioned and better clarified the central subject of our review, providing clarity and understanding to the reader.

Round 2

Reviewer 2 Report

Dear Authors,

Thank you for addressing all the suggestions and recommended corrections.

We found bit higher plagiarism report (19%) even after excluding the references. Rephrasing is highly recommended of lines 79-108; 207-242

Reviewer 3 Report

Accept in present form